# Distribution and predictive performance of the temporal phase of dynamic spot sign appearance in acute intracerebral hemorrhage

Hee Sahng Chung[1]ᵒ*, Santanu Chakraborty[2]ᵒ, Michael Reaume[1], Vignan Yogendrakumar[1], Matthew J. Hogan[1], Dylan Blacquiere[1], Grant Stotts[1], Michel Shamy[1], Richard I. Aviv[2], Dar Dowlatshahi[1]

**1** Department of Medicine, University of Ottawa, Ottawa, Canada, **2** Department of Medical Imaging, University of Ottawa, Ottawa, Canada

ᵒ These authors contributed equally to this work.

\* hchun030@uottawa.ca

**Data Availability Statement:** All relevant data are within the manuscript and its Supporting Information files.

## Abstract

### Background

Dynamic CT angiography (dCTA) contrast extravasation, known as the "dynamic spot sign", can predict hematoma expansion (HE) in intracerebral hemorrhage (ICH). Recent reports suggest the phase of spot sign appearance is related to the magnitude of HE. We used dCTA to explore the association between the phase of spot sign appearance and HE, clinical outcome, and contrast extravasation rates.

### Methods

We assessed consecutive patients who presented with primary ICH within 4.5 hours from symptom onset who underwent a standardized dCTA protocol and were spot sign positive. The independent variable was the phase of spot sign appearance. The primary outcome was significant HE (either 6 mL or 33% growth). Secondary outcomes included total absolute HE, mortality, and discharge mRS. Mann-Whitney U, Fisher's exact test, and logistic regression were used, as appropriate.

### Results

Of the 35 patients with spot signs, 27/35 (77%) appeared in the arterial phase and 8/35 (23%) appeared in the venous phase. Thirty patients had follow-up CT scans. Significant HE was seen in 14/23 (60.87%) and 3/7 (42.86%) of arterial and venous cohorts, respectively ($p = 0.67$). The sensitivity and specificity in predicting significant HE were 82% and 31% for the arterial phase and 18% and 69% for the venous phase, respectively. There was a non-significant trend towards greater total HE, in-hospital mortality, and discharge mRS of 4–6 in the arterial spot sign cohort. Arterial spot signs demonstrated a higher median contrast extravasation rate (0.137 mL/min) compared to venous spot signs (0.109 mL/min).

**Funding:** DD was supported by a Heart and Stroke Foundation of Canada Grant-in-Aid (https://www.heartandstroke.ca) and Clinician-Scientist Award from Department of Medicine, Ottawa Hospital (https://ottawadom.ca). RIA was supported by a Canadian Institutes of Health Research project grant in ICH (https://cihr-irsc.gc.ca/e/49051.html). The funders had no role in study design, data collection and analysis, decision to publish, or preparation of the manuscript.

**Competing interests:** Co-authors DD and MJH are authors of the Method and System for Identifying Bleeding patent (PAT 80049P-2 US). This does not alter our adherence to PLOS ONE policies on sharing data and materials.

## Conclusion

Our exploratory analyses suggest that spot sign appearance in the arterial phase may be more likely associated with HE and poorer prognosis in ICH. This may be related to higher extravasation rates of arterial phase spot signs. However, further studies with larger sample sizes are warranted to confirm the findings.

## Introduction

Hematoma expansion (HE) is a predictor of poor clinical outcome in intracerebral hemorrhage (ICH) [1]. Contrast extravasation on computed tomography angiography (CTA), known as the "spot sign", is a validated predictor for HE. However, conventional CTA is limited by its "static" acquisition of images [2].

Dynamic CTA (dCTA) acquires a series of CTA images over a 60-second period. We have previously reported that dCTA captures a higher prevalence of spot signs than conventional CTA [3]. However, the specificity of the spot sign on dCTA for predicting significant HE was lower than that of the spot sign on conventional CTA [3]. This lower specificity may be due to a lower rate of expansion associated with "delayed" spot signs (i.e. spot signs that appear in later phases of image acquisition). In support of this hypothesis, a recent study reported that spot signs captured in earlier phases of imaging acquisition on conventional CTA had stronger associations with HE compared to spot signs captured in delayed phases [4]. Therefore, we hypothesized that the phase of spot sign appearance on dCTA may be a novel predictor of HE and clinical prognosis. In the current study, the primary objective was to describe the performance of the arterial versus venous phase of spot sign appearance in predicting HE. Secondary objectives were to explore total absolute HE, rates of contrast extravasation as a surrogate of spot sign growth, mortality, and functional outcome across phases of spot sign appearance.

## Materials and methods

Ottawa Health Science Network Research Ethics Board (OHSN-REB) approved this study with a waiver of consent for data collection. We launched a prospective registry of consecutive patients aged $\geq 18$ years who presented to the Ottawa Hospital, Ontario, Canada, with spontaneous ICH within 4.5 hours from symptom onset. All patients with symptoms of stroke presenting under 4.5 hours received a dCTA as part of a standardized code stroke process. Patients with known secondary causes of ICH such as trauma, malignancy, aneurysms, or arteriovenous malformations were excluded.

The dCTA acquisition protocol has been previously described [5]. Briefly, we obtained non-contrast CT images followed by dCTA acquisition using 320-row volume CT scanner (Toshiba Aquilion ONE^TM). We acquired whole-brain angiographic images over a 60-second period: once at 7 seconds (used as a mask of subtraction) from the start of injection of contrast; then every 2 seconds from 10 to 35 seconds, followed by every 5 seconds to 60 seconds. An expert neuroradiologist (S.C.) blinded to follow-up scans and patient outcome determined the spot sign status [6]. We measured spot sign density using Quantomo Pro (Cybertrial, Calgary, AB). We estimated the rate of contrast extravasation as a measure of dCTA spot sign growth by calculating the slope of a time-density plot from the earliest point of spot sign appearance to the maximal contrast volume of the dCTA spot sign [3]. To classify the phase of spot sign appearance, we measured the maximum Hounsfield units (HU) of an arterial and venous

structure in the plane of the dynamic spot sign in the hemisphere contralateral to the ICH at time of spot sign appearance [4]. We then categorized the phase of spot sign appearance into five acquisition phases: early arterial, late arterial, equilibrium, peak venous, and late venous phases, as previously published [4]. We pooled the groups into two phases, arterial and venous, for data analyses.

The primary outcome was significant HE defined as an absolute growth of $\geq$ 6 mL or a relative increase of $\geq$ 33% in parenchymal hematoma volume on follow-up CT head [7]. Patients underwent a 24-hour follow-up CT as standard of care at our institution. We measured hematoma volumes using Quantomo Pro (Cybertrial, Calgary, AB). Secondary outcomes included median total HE defined as the absolute difference in total blood volume (ICH and IVH) between baseline and follow-up scans, in-hospital mortality and poor functional outcome (mRS 4–6) at discharge.

Sensitivity, specificity, positive predictive value (PPV), and negative predictive value (NPV) of arterial and venous phases in predicting significant HE were calculated with 95% confidence intervals. We evaluated radiologic and clinical outcomes across phases of spot sign appearance using Mann-Whitney U test or Fisher's exact test, as appropriate. We used logistic regression to assess phase of spot sign appearance as a predictor of significant HE. All statistics were performed using SPSS v23 (IBM, Armonk, NY). We applied complete case analysis for primary analysis, and extreme case analysis for sensitivity analysis. In the worst-case scenario, missing values for significant HE were assumed negative in the arterial group and positive in the venous group, and vice versa in the best-case scenario.

## Results

We enrolled 83 consecutive ICH patients, 5 of whom were excluded due to secondary causes of ICH. There were 35 spot positive patients, of whom 27 appeared in the arterial phase and 8 appeared in the venous phase. Thirty patients were included for HE analyses, and 35 patients were included for the remaining clinical outcome analyses. 4/27 patients in the arterial phase did not have follow-up CT scans due to palliation (n = 2) and early mortality (n = 2), and 1/8 in the venous phase did not have scans due to palliation (n = 1). Cohort selection flow diagram is presented in Fig 1. Baseline radiological characteristics for each cohort are presented in Table 1. No statistically significant differences in baseline characteristics were observed between the arterial and venous cohorts.

Phases of spot sign appearance are presented in Table 2. Spot signs appeared more frequently in the early arterial and peak arterial phases (62.9%, 14.3%) than in the peak venous and late venous phases (11.4%, 11.4%). Given that there were no spot signs appearing in the equilibrium phase, we pooled the early and peak arterial phases into an arterial cohort, and the peak and late venous phases into a venous cohort, respectively. Higher proportions of significant HE were observed in the arterial phase of spot sign appearance, but these results were not statistically significant (P = 0.67).

The diagnostic accuracy of the phase of spot sign appearance in predicting significant HE is summarized in Table 3. Arterial phase spot sign appearance demonstrated higher sensitivity but lower specificity in predicting significant HE than venous phase. Primary and secondary outcomes are presented in Table 4. Overall, we observed a non-significant trend towards greater proportions of significant HE, in-hospital mortality, poor neurologic status (mRS 4–6) and higher total HE in the arterial cohort. Worst-case and best-case scenarios for sensitivity analyses did not reveal any statistically significant difference in the proportions of significant HE between the arterial and venous phase cohorts (P = 1.00 and P = 0.22, respectively).

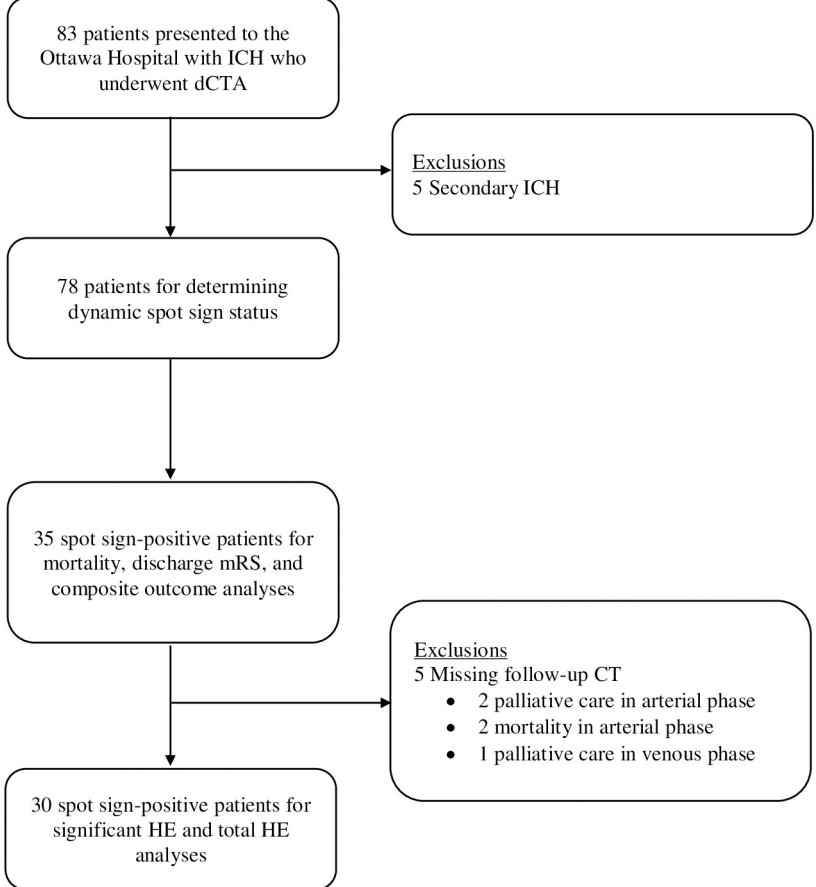

**Fig 1. Cohort selection flow diagram.** ICH = intracerebral hemorrhage. dCTA = dynamic computed tomography angiogram. ICH = intracerebral hemorrhage. mRS = modified Rankin scale. CT = computed tomography. HE = hematoma expansion.

Dynamic CTA allowed assessment of contrast extravasation between start and end points of spot sign appearance. We demonstrate the evolution of spot sign formation in arterial phase (S1 Video) and in venous phase (S2 Video). The median rates of spot sign growth were 0.137 (IQR 0.081–0.399) mL/min and 0.109 (IQR 0.033–0.171) mL/min in the arterial and venous cohorts, respectively.

## Discussion

Our findings suggest that dCTA spot signs that appear in the arterial phase have a higher sensitivity but lower specificity to predict HE compared to those that appear in the venous phase. This is consistent with a recent report using conventional CTA acquired in different phases [4]. Our study provides further support for this concept by demonstrating a higher rate of contrast extravasation in the arterial phase.

There was an overall non-significant trend towards higher frequencies of significant HE, mortality, discharge mRS 4–6, and greater total HE in the arterial cohort of spot sign appearance. These findings suggest that spot sign appearance in the arterial phase may have higher correlations with significant HE and poor clinical prognosis, but that our study was underpowered to detect statistically significant differences. This is consistent with earlier studies that used repeat CTA or CTA followed by post contrast CT to detect a greater number of spot signs

**Table 1. Cohort characteristics.**

| | Grouped Arterial Phase (Early + Peak) | Grouped Venous Phase (Peak + Late) | P Value |
|---|---|---|---|
| **Total N** | 27 | 8 | |
| **Demographics** | | | |
| Age (years) | 79 (69.5, 84) | 82.5 (72.25, 88.25) | 0.56 |
| Male | 11/27 (40.74) | 4/8 (50) | 0.7 |
| **PMHx** | | | |
| Ischemic Stroke or TIA | 6/27 (22.22) | 1/8 (12.5) | 1 |
| Coronary Artery Disease | 2/27 (7.41) | 0/8 (0) | 1 |
| PE/DVT | 4/27 (14.81) | 0/8 (0) | 0.55 |
| Intracerebral Hemorrhage | 2/27 (7.41) | 1/8 (12.5) | 0.55 |
| Subarachnoid Hemorrhage | 2/27 (7.41) | 0/8 (0) | 1 |
| Subdural Hemorrhage | 1/27 (3.7) | 0/8 (0) | 1 |
| Hypertension | 14/27 (51.85) | 4/8 (50) | 1 |
| Diabetes Mellitus | 5/27 (18.52) | 1/8 (12.5) | 1 |
| Atrial Fibrillation | 4/27 (14.81) | 2/8 (25) | 0.6 |
| Congestive Heart Failure | 2/27 (7.41) | 2/8 (25) | 0.22 |
| Current Smoker | 2/27 (7.41) | 1/8 (12.5) | 0.55 |
| **Medication Hx** | | | |
| Aspirin | 8/27 (29.63) | 2/8 (25) | 1 |
| Plavix | 3/27 (11.11) | 1/8 (12.5) | 1 |
| Warfarin | 4/27 (14.81) | 1/8 (12.5) | 1 |
| Heparin | 2/27 (7.41) | 0/8 (0) | 1 |
| **Baseline Clinical** | | | |
| Premorbid mRS Score | 0 (0, 1.5) | 0 (0, 2.5) | 0.59 |
| GCS Score[a] | 14 (11, 15) | 14 (12.5, 15) | 0.68 |
| NIHSS Score[b] | 14 (10, 18.75) | 12 (9.5, 17.5) | 0.49 |
| Systolic blood pressure (mmHg) | 178 (160.5, 187.5) | 177.5 (162.25, 198.75) | 1 |
| Diastolic blood pressure (mmHg)[c] | 90 (78, 102.75) | 94 (82, 100) | 0.94 |
| Glucose (mmol/L) | 6.6 (6.05, 9.3) | 6.55 (6.28, 7.33) | 0.86 |
| Platelets (x$10^3$ cells / L) | 192 (162, 251) | 177.5 (158, 197.25) | 0.29 |
| INR[d] | 1 (1, 1.2) | 1.2 (1.05, 1.5) | 0.19 |
| Creatinine (μmol/L) | 77 (62.5, 95.5) | 76 (65.25, 90) | 0.95 |
| **Baseline Radiologic** | | | |
| ICH Volume Baseline | 35.53 (20.41, 58.25) | 21.81 (13.24, 51.64) | 0.53 |
| IVH Volume Baseline | 0 (0, 0.79) | 0.9 (0, 2.91) | 0.29 |
| Total Volume Baseline | 37.62 (24.91, 58.25) | 23.62 (14.16, 66.35) | 0.61 |
| **Process Measures** | | | |
| Time from Onset to Baseline CT/CTA (min) | 74 (62, 90) | 75 (43.75, 101.75) | 0.64 |
| Time between Baseline CT and Follow-Up CT (min)[e] | 16.4 (9.3, 25.08) | 19.7 (13.88, 24.88) | 0.75 |
| Total Hematoma Volume / Time from Onset to Baseline CT, CTA (mL/min) | 0.51 (0.3, 0.71) | 0.38 (0.2, 0.85) | 0.84 |

Data are n/N (%) or median (25th, 75th percentile).

a: N = 26 for arterial phase due to 1 missing value, N = 7 for venous phase due to 1 missing value.

b: N = 22 for arterial phase due to 5 missing values, N = 7 for venous phase due to 1 missing value.

c: N = 26 for arterial phase due to 1 missing value, N = 8 for venous phase.

d: N = 27 for arterial phase, N = 7 for venous phase due to 1 missing value.

e: N = 23 for arterial phase due to 4 missing follow-up CT, N = 7 for venous phase due to 1 missing follow-up CT

TIA = transient ischemic attack, PE = pulmonary embolism, DVT = deep vein thrombosis, mRS = modified Rankin Scale, GCS = Glasgow Coma Scale, NIHSS = National Institutes of Health Stroke Scale, INR = international normalized ratio, ICH = intracerebral hemorrhage, IVH = intraventricular hemorrhage, CTA = CT angiography.

**Table 2. Significant HE across phases of spot sign appearance.**

|  | Grouped Arterial Phase | Grouped Venous Phase |
|---|---|---|
| **Spot Sign Appearance[a]** | 27/35 (77.1) | 8/35 (22.9) |
| **Significant HE[b]** | 14/23 (60.9) | 3/7 (42.9) |

a: Data expressed as N of spot sign appearance in each phase / Total N of spot signs (%).

b: Significant HE defined as ICH growth ≥ 6 mL or ≥ 33% from baseline. Data expressed as N of significant HE / N of spot signs in each phase with follow-up CT (%). Phase determined by spot sign appearance.

**Table 3. Diagnostic accuracy of phase of spot sign appearance in predicting significant HE.**

|  | Grouped Arterial Phase % (95% CI) | Grouped Venous Phase % (95% CI) |
|---|---|---|
| **Sensitivity** | 82 (57–96) | 18 (4–43) |
| **Specificity** | 31 (9–61) | 69 (39–91) |
| **Positive Predictive Value** | 61 (50–70) | 43 (17–74) |
| **Negative Predictive Value** | 57 (26–83) | 39 (30–50) |

**Table 4. Radiologic and clinical outcomes across phases of spot sign appearance.**

|  | Grouped Arterial Phase | Grouped Venous Phase | P Value |
|---|---|---|---|
| **Spot Signs[a]** | 27/35 (77.1) | 8/35 (22.9) |  |
| **Significant HE[b]** | 14/23 (60.9) | 3/7 (42.9) | 0.67 |
| **Median Total HE[c]** | 10.02 (2.8, 24.1) | 4.35 (0.8, 26.6) | 0.77 |
| **Mortality[d]** | 13/27 (48.2) | 2/8 (25.0) | 0.42 |
| **Discharge mRS 4–6[e]** | 25/27 (92.6) | 5/8 (62.5) | 0.07 |

a: Data expressed as N of spot signs in each phase / Total N of spot signs (%).

b: Significant HE defined as ICH growth ≥ 6 mL or ≥ 33% from baseline. Data expressed as N of significant HE / N of spot signs in each cohort with follow-up CT (%).

c: Total HE defined as the sum of ICH and IVH expansion from baseline. Data expressed as mL (25th, 75th percentiles).

d: Data expressed as N of in-hospital mortality / N of spot signs in each cohort (%).

e: Data expressed as N of discharge mRS 4–6 / N of spot signs in each cohort (%).

f: Data expressed as N of mortality or significant HE / N of spot signs in each cohort (%).

and assess risk of hematoma expansion. Ederies et al. showed that CTA-detected spot signs were more likely associated with HE and poor outcome than post-contrast CT spot signs [8]. D'Esterre et al. quantified the contrast leakage rates between these different groups, demonstrating that CTA-detected spot signs had higher contrast leakage rates than post-contrast CT spot signs [9]. Brouwers et al. reported that higher spot sign extravasation rates on multiphase CTA were independently associated with greater HE and poorer prognosis [10]. Indeed, we observed higher rates of spot sign growth among spot signs appearing in the arterial phase. Altogether, the data suggest that spot signs appearing in the arterial phase may represent faster bleeding, leading to larger HE.

This study is largely exploratory and has limitations. The sample size is small and underpowered to fully evaluate the relationship between the phase of spot sign appearance and clinical outcomes. Furthermore, 15% (4/27) patients were lost to follow-up in the arterial cohort, and 13% (1/8) in the venous cohort. However, our sensitivity analyses did not reveal a change

in results. Finally, the calculation of extravasation rates in our study assumes a constant linear relationship between the spot sign volume and the timespan of image acquisition. Emerging automated machine learning measurement techniques may better model extravasation rates in future studies.

## Conclusion

The predictability of significant HE and clinical outcome may differ across the timing of spot sign appearance on dCTA. Our results suggest that the majority of spot signs appear in the arterial phase, which has a higher sensitivity to predict HE and poor outcome in ICH. This may be related to higher extravasation rates of arterial phase spot signs, reflecting faster bleeding. Further studies with larger sample sizes are needed to confirm the findings of our study and to better explore the relationship between phase and rate of extravasation.

## Supporting information

**S1 Data.**
(XLSX)

**S1 Video. Dynamic CT angiogram shows temporal evolution of a spot sign that appears in arterial phase in a subject with a large right intraparenchymal frontal hematoma.** The spot sign is located at the 9 o'clock position indicated by the arrow.
(MP4)

**S2 Video. Dynamic CT angiogram shows temporal evolution of a spot sign that appears in venous phase in a subject with a large right intraparenchymal parietal hematoma.** The spot sign is located at the 6 o'clock position indicated by the arrow.
(MP4)

## Author Contributions

**Conceptualization:** Santanu Chakraborty, Vignan Yogendrakumar, Matthew J. Hogan, Richard I. Aviv, Dar Dowlatshahi.

**Data curation:** Hee Sahng Chung, Santanu Chakraborty, Michael Reaume, Matthew J. Hogan, Dar Dowlatshahi.

**Formal analysis:** Hee Sahng Chung, Dar Dowlatshahi.

**Funding acquisition:** Richard I. Aviv, Dar Dowlatshahi.

**Investigation:** Hee Sahng Chung, Santanu Chakraborty, Michael Reaume, Dar Dowlatshahi.

**Methodology:** Hee Sahng Chung, Santanu Chakraborty, Vignan Yogendrakumar, Dylan Blacquiere, Grant Stotts, Michel Shamy, Richard I. Aviv, Dar Dowlatshahi.

**Project administration:** Santanu Chakraborty, Dar Dowlatshahi.

**Supervision:** Santanu Chakraborty, Richard I. Aviv, Dar Dowlatshahi.

**Validation:** Vignan Yogendrakumar, Matthew J. Hogan, Dylan Blacquiere, Grant Stotts, Michel Shamy, Richard I. Aviv, Dar Dowlatshahi.

**Visualization:** Santanu Chakraborty, Vignan Yogendrakumar, Matthew J. Hogan, Richard I. Aviv, Dar Dowlatshahi.

**Writing – original draft:** Hee Sahng Chung, Santanu Chakraborty, Dar Dowlatshahi.

**Writing – review & editing:** Hee Sahng Chung, Santanu Chakraborty, Michael Reaume, Vignan Yogendrakumar, Matthew J. Hogan, Dylan Blacquiere, Grant Stotts, Michel Shamy, Richard I. Aviv, Dar Dowlatshahi.

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
