## [Decision Letter · Decision Letter 0]

16 Jun 2020

PONE-D-20-11565

Distribution and predictive performance of the temporal phase of dynamic spot sign appearance in acute intracerebral hemorrhage

PLOS ONE

Dear Dr. Chung,

Thank you for submitting your manuscript to PLOS ONE. After careful consideration, we feel that it has merit but does not fully meet PLOS ONE’s publication criteria as it currently stands. Therefore, we invite you to submit a revised version of the manuscript that addresses the points raised during the review process.

We look forward to receiving your revised manuscript.

Kind regards,

Jonathan H Sherman

Academic Editor

PLOS ONE

Journal Requirements:

2. We note that you have a patent relating to material pertinent to this article. Please provide an amended statement of Competing Interests to declare this patent (with details including name and number), along with any other relevant declarations relating to employment, consultancy, patents, products in development or modified products etc. Please confirm that this does not alter your adherence to all PLOS ONE policies on sharing data and materials, as detailed online in our guide for authors http://journals.plos.org/plosone/s/competing-interests by including the following statement: "This does not alter our adherence to  PLOS ONE policies on sharing data and materials.” If there are restrictions on sharing of data and/or materials, please state these. Please note that we cannot proceed with consideration of your article until this information has been declared.

Additional Editor Comments (if provided):

Reviewers' comments:

Reviewer's Responses to Questions

**Comments to the Author**

1. Is the manuscript technically sound, and do the data support the conclusions?

Reviewer #1: Yes

Reviewer #2: Yes

2. Has the statistical analysis been performed appropriately and rigorously? 

Reviewer #1: Yes

Reviewer #2: Yes

3. Have the authors made all data underlying the findings in their manuscript fully available?

Reviewer #1: Yes

Reviewer #2: Yes

4. Is the manuscript presented in an intelligible fashion and written in standard English?

Reviewer #1: Yes

Reviewer #2: Yes

5. Review Comments to the Author

Reviewer #1: The authors examined 'spot sign' on dynamic CTA's (arterial vs. venous phase) ability to predict significant hematoma expansion (defined as 6mL or 33% relative volume growth). This is an important topic as clinical predictors will likely be of value when deciding on trial enrollment for potential therapeutics for ICH. The manuscript is well-written, direct, and easy to follow. The intent of the study is clearly stated and the methods for which the authors attempt to answer their hypothesis is rationale and scientifically sound.

The major limitation of the study is the sample size in each group, in particular the venous phase (n=8), which the authors explicitly mention in the discussion. With this low of a sample size it is challenging to draw a firm conclusion, however, the authors appropriately conclude that the study is 'exploratory' and 'predictability of significant HE and clinical outcome MAY differ across the timing of spot sign appearance on dCTA' and 'larger sample sizes are needed to confirm the findings of our study.' The authors collected subjects prospectively but could consider including retrospective cases if dCTA was being used at their center before they started the study to increase the number of subjects.

A few minor requests:

MEdian time of follow-up CT is listed in the Table however please comment on your sites/study's customary practice for serial neuroimaging. For example, repeat CT in all cases within xx h, repeat only if clinically worse.

Please describe method used to measure hematoma volume (automated segmentation, abc2, etc)

Reviewer #2: This article demonstrating a trend towards poor outcome in ICH patients with arterial phase spot sign on dCTA when compared to venous cohorts is well received. The small sample size is mentioned, and publication of this article will highlight the need for further study.

6. PLOS authors have the option to publish the peer review history of their article (what does this mean?). If published, this will include your full peer review and any attached files.

Reviewer #1: No

Reviewer #2: Yes: Kathleen Burger DO

---

## [Author Response · Author response to Decision Letter 0]

29 Jun 2020

Author response: Thank you, this has been corrected. Changes are trackable in our revised manuscript file.

2. We note that you have a patent relating to material pertinent to this article. Please provide an amended statement of Competing Interests to declare this patent (with details including name and number), along with any other relevant declarations relating to employment, consultancy, patents, products in development or modified products etc. Please confirm that this does not alter your adherence to all PLOS ONE policies on sharing data and materials, as detailed online in our guide for authors http://journals.plos.org/plosone/s/competing-interests by including the following statement: "This does not alter our adherence to PLOS ONE policies on sharing data and materials.” If there are restrictions on sharing of data and/or materials, please state these. Please note that we cannot proceed with consideration of your article until this information has been declared.

Author response: Name and number are included. I confirm this does not alter our adherence to all PLOS ONE policies on sharing data and materials and will add this statement. I have included the disclosure information in our cover letter.

Additional Editor Comments (if provided):

5. Review Comments to the Author

Reviewer #1: The authors examined 'spot sign' on dynamic CTA's (arterial vs. venous phase) ability to predict significant hematoma expansion (defined as 6mL or 33% relative volume growth). This is an important topic as clinical predictors will likely be of value when deciding on trial enrollment for potential therapeutics for ICH. The manuscript is well-written, direct, and easy to follow. The intent of the study is clearly stated and the methods for which the authors attempt to answer their hypothesis is rationale and scientifically sound.

The major limitation of the study is the sample size in each group, in particular the venous phase (n=8), which the authors explicitly mention in the discussion. With this low of a sample size it is challenging to draw a firm conclusion, however, the authors appropriately conclude that the study is 'exploratory' and 'predictability of significant HE and clinical outcome MAY differ across the timing of spot sign appearance on dCTA' and 'larger sample sizes are needed to confirm the findings of our study.' The authors collected subjects prospectively but could consider including retrospective cases if dCTA was being used at their center before they started the study to increase the number of subjects.

Author response: Thank you for your feedback and suggestion. Because our study was prospective, our REB approval was to only collect the data under the protocol funded by the Heart & Stroke Foundation of Canada. It is unlikely they will allow retrospective data collection outside the original protocol, and would require a full new submission and review. While this is not impossible, it would take considerable time for the review (which is often 6 months at our institution). Furthermore, due to COVID, non-COVID submissions are being triaged as low priority, so we are concerned this would further delay reviews.

A few minor requests:

Median time of follow-up CT is listed in the Table however please comment on your sites/study's customary practice for serial neuroimaging. For example, repeat CT in all cases within xx h, repeat only if clinically worse.

Author response: Thank you for pointing this out. This has been included in lines 134-135.

Please describe method used to measure hematoma volume (automated segmentation, abc2, etc)

Author response: Thank you, this has been included in lines 135-136.

Reviewer #2: This article demonstrating a trend towards poor outcome in ICH patients with arterial phase spot sign on dCTA when compared to venous cohorts is well received. The small sample size is mentioned, and publication of this article will highlight the need for further study.

Author response: We thank the reviewer.

---

## [Decision Letter · Decision Letter 1]

1 Jul 2020

Distribution and predictive performance of the temporal phase of dynamic spot sign appearance in acute intracerebral hemorrhage

PONE-D-20-11565R1

Dear Dr. Chung,

We’re pleased to inform you that your manuscript has been judged scientifically suitable for publication and will be formally accepted for publication once it meets all outstanding technical requirements.

Kind regards,

Jonathan H Sherman

Academic Editor

PLOS ONE

Additional Editor Comments (optional):

Reviewers' comments:

Reviewer's Responses to Questions

**Comments to the Author**

1. If the authors have adequately addressed your comments raised in a previous round of review and you feel that this manuscript is now acceptable for publication, you may indicate that here to bypass the “Comments to the Author” section, enter your conflict of interest statement in the “Confidential to Editor” section, and submit your "Accept" recommendation.

Reviewer #1: All comments have been addressed

2. Is the manuscript technically sound, and do the data support the conclusions?

Reviewer #1: Yes

3. Has the statistical analysis been performed appropriately and rigorously? 

Reviewer #1: Yes

4. Have the authors made all data underlying the findings in their manuscript fully available?

Reviewer #1: Yes

5. Is the manuscript presented in an intelligible fashion and written in standard English?

Reviewer #1: Yes

6. Review Comments to the Author

Reviewer #1: The authors addressed all questions. The reviewer acknowledges the inability of the author to gather retrospective data and does not feel this limitation hinders the work to a point that it should not be shared/published. No further remarks apart from this being a well written manuscript.

7. PLOS authors have the option to publish the peer review history of their article (what does this mean?). If published, this will include your full peer review and any attached files.

Reviewer #1: No